# Anti-Aging Effect of *Hemerocallis citrina* Baroni Polysaccharide-Rich Extract on *Caenorhabditis elegans*

**DOI:** 10.3390/ijms25010655

**Published:** 2024-01-04

**Authors:** Yunxia Zou, Xiyue Qin, Wenli Wang, Qingyong Meng, Yali Zhang

**Affiliations:** 1School of Food Science and Nutritional Engineering, East Campus, China Agricultural University, Beijing 100083, China; zouyunxia@cau.edu.cn (Y.Z.); qxy1805@163.com (X.Q.); wenliwang@cau.edu.cn (W.W.); 2School of Biology, West Campus, China Agricultural University, Beijing 100193, China; qymeng@cau.edu.cn

**Keywords:** *Hemerocallis citrina* Baroni, polysaccharide, *C. elegans*, antioxidant, longevity

## Abstract

Plant polysaccharides are important for anti-aging research. Polysaccharides from *Hemerocallis citrina* Baroni (*H. citrina*) have been reported to have antioxidant activity; however, their anti-aging roles and mechanisms are not clear. In this study, we extracted polysaccharides from *H. citrina* by an ultrasonic-assisted water extraction–alcohol precipitation method and chemically determined the physicochemical properties such as extraction yield, content, and in vitro antioxidant properties of *H. citrina* polysaccharide-rich extract (HCPRE). Using *Caenorhabditis elegans* (*C. elegans*) as a model animal, the anti-aging effect of HCPRE was investigated, and the mechanism of action of HCPRE was explored by the in vivo antioxidant level assay of *C. elegans* and the related gene expression assay. The extraction yield of HCPRE was 11.26%, the total polysaccharide content was 77.96%, and the main monosaccharide components were glucose and galactose. In addition, HCPRE exhibited good antioxidant activity both in vitro and in vivo. Under normal thermal stress and oxidative stress conditions, being fed 1200 µg/mL of HCPRE significantly prolonged the life span of *C. elegans* by 32.65%, 17.71%, and 32.59%, respectively. Our study showed that HCPRE exerted an anti-aging effect on *C. elegans*, and its mechanism involves increasing the activities of catalase (CAT) and superoxide dismutase (SOD), reducing the level of reactive oxygen species (ROS) and regulating the expression of related genes.

## 1. Introduction

Aging is an irreversible degenerative change in structure and function that occurs gradually in all tissues and organs of an organism with age under the influence of various internal and external factors such as genetics, mental stress, and environmental pollution [1]. Many researchers have proposed theories about aging, such as the neuroendocrine, immune, free radical, telomere, and metabolic waste accumulation theories [2,3]. The free radical theory of aging is now supported by evidence. This theory states that the body’s antioxidant system contains a large number of antioxidants and antioxidant enzymes and that the generation of antioxidant plasmids and reactive oxygen radicals in the body are present in a dynamic balance [4]. In recent years, important progress has been made in the research on natural plants and their potential in the development of new drugs and other aspects due to the high efficiency and low toxicity of their biologically active substances having been recognized. Plant polysaccharides are of great significance in the study of anti-aging, and currently, an increasing number of studies focus on the anti-aging activity of plant polysaccharides [5,6,7].

*Hemerocallis citrina* Baroni (*H. citrina*), also known as yellow flower vegetable, daylilies, and forget-me-nots, is a perennial herb in the genus *Hemerocallis* in the subfamily of *Asphodeloideae* of the family Liliaceae, mainly cultivated in China and Japan [8]. *H. citrina* production is expanding rapidly; China ranks first worldwide, with an annual output of 580,000 t and approximately 304,000 hm^2^, accounting for 97% of the world’s total planted area [9]. In traditional Chinese medicine, *H. citrina* is a therapeutic food with several beneficial properties. It is rich in nutrients, including vitamins, sugars, amino acids, proteins, and inorganic salts, and has antidepressant, lipid-lowering, hypoglycemic, antitumor, antioxidant, and antibacterial activities [10,11,12,13,14,15]. In one study, extracts of *H. citrina* rich in polysaccharides were shown to be scavengers of superoxide anion and hydroxyl radicals [16]. In addition, other studies have shown that *H. citrina* polysaccharides have strong 2,2′-azinobi-(3-ethylbenzthiazoline-6-ulphonate) radical (ABTS-scavenging capacity and ferric-reducing antioxidant power (FRAP) [17,18]. Thus, *H. citrina* polysaccharide may be an effective anti-aging agent; however, more research is required.

*C. elegans* genes are homologous to 80% of human genes. The species is easy to observe and has a short experimental period; thus, it is often used in anti-aging research and can also be used to screen natural products with anti-aging activity [19]. Adults of *C. elegans* have a body length of 1.0–1.5 mm, a diameter of approximately 70 µm, are translucent, feed mainly on *Escherichia coli* (*E. coli*), and in the natural state are mostly hermaphroditic nematodes, laying approximately 300 eggs during their lifetime by autotrophic fertilization. t takes only about 3 days for *C. elegans* to develop from a fertilized egg through the embryonic and larval stages to an adult at 20 °C. The larval stage is divided into four stages: L1, L2, L3, and L4 [20].

This study aimed to investigate the anti-aging effect and mechanism of action of *H. citrina* polysaccharide-rich extract (HCPRE). We used the ultrasonic-assisted water extraction–alcohol precipitation method to extract *H. citrina* polysaccharide, and the physicochemical properties such as the extraction yield and content of HCPRE were determined. The molecular weight, monosaccharide composition, and in vitro antioxidant properties were determined and analyzed. *C. elegans* was used as a model animal and fed different concentrations of HCPRE to investigate the anti-aging role of HCPRE in *C. elegans* and to explore the mechanism of action, which is of great importance for the comprehensive development and utilization of *H. citrina* resources [21].

## 2. Results

### 2.1. Extraction and Component Analysis of HCPRE

The HCPRE extraction yield was calculated to be 11.26%, the ratio of polysaccharides in HCPRE was 77.96%, the ratio of glyoxylates was 2.62%, the ratio of total phenolic acids was 1.21%, and the ratio of proteins was 0.85%, and it did not contain starch or reducing sugars. The molecular weight distribution of HCPRE (Figure 1) showed that there were three fractions comprising 921,977 Da, 2308 Da, and 200 Da. These accounted for 37.636%, 45.376%, and 16.986% of HCPRE, respectively. The results of the monosaccharide fractions of HCPRE (Figure 2) showed that the monosaccharides contained in HCPRE were mannose, ribose, rhamnose, glucuronide, galacturonic acid, glucose, galactose, xylose, arabinose, and fucose; the relative contents of these are shown in Table 1. Among them, glucose had the highest content (31.43%), and glucuronic acid had the lowest content (0.36%).

### 2.2. Antioxidant Properties of HCPRE In Vitro

The results of the DPPH radical scavenging experiments using HCPRE are shown in Figure 3A. The DPPH radical scavenging activity of HCPRE increased gradually with increasing concentration of HCPRE in the concentration range examined and showed a concentration-dependent relationship. At concentrations ≥400 µg/mL, HCPRE showed significant DPPH radical scavenging activity compared to glucose. The total reduction capacity of HCPRE is shown in Figure 3B. As shown in the figure, the total reducing power of HCPRE was enhanced with increasing sample concentration in the concentration range examined and showed a concentration-dependent relationship. At concentrations ≥600 µg/mL, HCPRE had a significantly higher total reducing power compared to that of glucose. In conclusion, HCPRE exhibited good antioxidant properties in vitro.

### 2.3. Effects of HCPRE on Body Length, Spawning Capacity, and Longevity of C. elegans

The length of *C. elegans* after being fed different HCPRE concentrations is shown in Figure 4A. The first three days present the rapid growth period and days 4–5 the stabilization period, when the length of *C. elegans* did not change significantly. Day 6 was the senescence period when body length began to decrease. The body length of *C. elegans* reflects its degree of growth and development. The body length of *C. elegans* becomes significantly shorter in unfavorable environments. During the experimental period, HCPRE within the experimental concentration range did not impair the growth and development of *C. elegans* under normal conditions.

As shown in Figure 4B and Table 2, the survival rate of *C. elegans* gradually decreased with an increase in the number of days since birth, and a substantial decrease was observed on day 13. The average life span of the blank control group was 10.78 days, which increased with increasing HCPRE concentration. After feeding with 800 µg/mL HCPRE, the average life span of *C. elegans* was 13.32 days, which was 23.56% higher than the blank control group. After feeding with 1200 µg/mL HCPRE, *C. elegans* had the highest mean life span of 14.30 days, which was 32.65% higher than the blank control group. This suggests that HCPRE in the range of 800 µg/mL to 1200 µg/mL had a prolonged effect on the mean life span of both *C. elegans*.

The offspring are shown in Figure 4C,D. As shown in Figure 4C, the offspring of *C. elegans* fed different concentrations of HCPRE showed a tendency to increase and then decrease over time during the spawning period from day 2 to 6, and the peak of spawning was mainly concentrated on days 3 and 4. As shown in Figure 4D, feeding with 400, 800, and 1200 µg/mL of HCPRE can significantly increase the total number of offspring of *C. elegans* compared with the blank control group. Among them, feeding with 1200 µg/mL HCPRE was more effective. This suggests that HCPRE in the experimental concentration range improves the reproductive ability of *C. elegans*.

### 2.4. Effects of HCPRE on C. elegans under Acute Stress

The life span of *C. elegans* after being fed different HCPRE concentrations under acute thermal stress are shown in Table 3 and Figure 5. The mean life span of the blank control group was 7.17 h. The highest mean life span of *C. elegans* was 8.44 h after being fed 1200 µg/mL HCPRE, which was 17.71% higher compared to that of the blank control group. This suggests that being fed 1200 µg/mL HCPRE under acute heat stress has a significant life span extension effect on *C. elegans*.

The life span of *C. elegans* after being fed different HCPRE concentrations under acute oxidative stress are shown in Table 4 and Figure 6. The mean life span of the blank control group was 4.51 h. The highest mean life span of *C. elegans* was 5.98 h after being fed 1200 µg/mL HCPRE, which was 32.59% higher compared to that in the blank control group. This suggests that being fed 1200 µg/mL HCPRE under acute oxidative stress has a significant effect on extending the life span of *C. elegans*.

### 2.5. Effect of HCPRE on Antioxidant Properties of C. elegans In Vivo

The natural morphology of *C. elegans* in the blank control group without added HCPRE at each period is shown in Figure 7. On day 4, *C. elegans* had just entered the adult stage; its body was the most flexible, and all the physiological indices were in the best condition. Figure 7 shows a clear internal organ structure and smooth epidermis, which represents the pre-growth stage. On day 8, the physiological indicators of *C. elegans* began to decline, but the epidermis remained intact, indicating a mid-growth stage. On day 16, *C. elegans* was approaching its maximum life span; its body was almost no longer oscillating, its movement was slow, and the epidermal damage was unclear, representing the late stage of *C. elegans* growth. Therefore, we chose *C. elegans* at days 4, 8, and 16 to investigate the effect of HCPRE on the antioxidant properties.

CAT is an important antioxidant enzyme that reduces oxidative stress by decomposing hydrogen peroxide into water and oxygen [22]. CAT activity in *C. elegans* is shown in Figure 8A, which exhibits a tendency to increase and then decrease with time, with peaks mainly concentrated at around day 8. Being fed 800 and 1200 µg/mL HCPRE significantly increases the in vivo CAT activity in *C. elegans* on days 4, 8, and 16. The in vivo CAT activity of *C. elegans* fed 1200 µg/mL HCPRE increased the most.

SOD is a widely used antioxidant enzyme that converts superoxide radicals into hydrogen peroxide, thereby protecting organisms in oxygenated environments [23]. The activity of SOD in *C. elegans* is shown in Figure 8B, exhibiting a trend of increasing and then decreasing with time, with the peak again mainly centered around day 8. In contrast, being fed 1200 µg/mL HCPRE significantly increased SOD activity in *C. elegans* on days 4 and 8, and the effect was more pronounced on day 4. On day 16, there was no significant difference between *C. elegans* fed different concentrations of HCPRE and the blank control group.

Under normal conditions, ROS is maintained in a dynamic balance in organisms; however, an excess of ROS can lead to damage to the organism [24,25]. The levels of ROS in *C. elegans* are shown in Figure 8C; they gradually increased with the aging of *C. elegans*. In the middle and late stages of *C. elegans* growth, the in vivo ROS levels in the experimental group were significantly lower than those in the blank control group, with the most significant effect in the 1200 µg/mL HCPRE-treated group. This indicates that HCPRE can reduce the ROS level in *C. elegans* to a certain extent during the growth process and prevent oxidative damage, thus prolonging the life span. Combined with the CAT and SOD results, these results suggest that HCPRE enhances the antioxidant capacity of *C. elegans*.

### 2.6. Effects of HCPRE on Gene Expression in C. elegans In Vivo

#### 2.6.1. Effect of HCPRE on the Expression of Longevity-Related Genes in *C. elegans*

As shown in Figure 9, the expression of *daf-2* and *age-1* genes in *C. elegans* in the blank control group gradually increased over time. Only being fed 1200 µg/mL HCPRE could significantly reduce the gene expression of *daf-2* and *age-1* at day 4.

#### 2.6.2. Effect of HCPRE on the Expression of CAT-Related Genes in *C. elegans*

As shown in Figure 10A, *ctl-1* expression in *C. elegans* in the blank control group gradually increased over time. Compared with the blank control group, the expression of the *ctl-1* gene in *C. elegans* fed 1200 µg/mL HBCP on day 4 was significantly increased. Meanwhile, the expression of the *ctl-1* gene in *C. elegans* fed 1200 µg/mL HBCP was significantly higher than that in *C. elegans* fed other concentrations of HCPRE on days 4, 8, and 16. As shown in Figure 10B, the expression of the *ctl-2* gene in *C. elegans* in the blank control group showed a tendency to first increase and then decrease over time. Compared with the blank control group, *ctl-2* gene expression in *C. elegans* fed 800 µg/mL HCPRE was significantly increased on days 4 and 8, and *ctl-2* gene expression in *C. elegans* fed 1200 µg/mL HCPRE was significantly increased only on day 4.

#### 2.6.3. Effect of HCPRE on the Expression of SOD-Related Genes in *C. elegans*

As shown in Figure 11A, *sod-1* expression in *C. elegans* in the blank control group gradually increased over time. Compared with the blank control group, the expression of the *sod-1* gene in *C. elegans* fed 1200 µg/mL HCPRE was significantly increased on days 4 and 16. As shown in Figure 11B, the expression of the *sod-2* gene in *C. elegans* in the blank control group also gradually increased over time. Compared with the blank control group, the expression of the *sod-2* gene in *C. elegans* fed 1200 µg/mL HCPRE increased significantly on day 8. As shown in Figure 11C, the expression of the *sod-3* gene in *C. elegans* in the blank control group tended to increase and then decrease with time. On day 4, the expression of the *sod-3* gene in *C. elegans* fed 1200 µg/mL HCPRE was significantly higher than that in the blank control group. On day 8, the expression of the *sod-3* gene in *C. elegans* fed different concentrations of HCPRE was significantly higher than that in the blank control group. As can be seen from Figure 11D, the expression of the *sod-5* gene in *C. elegans* in the blank control group gradually decreased over time. Compared with the blank control group, the expression of the *sod-5* gene in *C. elegans* fed different concentrations of HCPRE was significantly higher on day 16.

#### 2.6.4. Effect of HCPRE on the Expression of Thermal-Stress-Related Genes in *C. elegans*

As shown in Figure 12A, *hsp-60* expression in *C. elegans* in the blank control group gradually increased over time. On day 8, the expression of the *hsp-60* gene in *C. elegans* fed 1200 µg/mL HCPRE was significantly higher than that in the blank control group. As shown in Figure 12B,C, the expression of both the *hsp-16.1* gene and the *hsp-16.2* gene in *C. elegans* in the blank control group showed an increasing trend and then decreased over time. On day 8, the expression of both the *hsp-16.1* gene and the *hsp-16.2* gene in *C. elegans* fed 800 and 1200 µg/mL HCPRE was significantly higher than that in the blank control group.

#### 2.6.5. Effect of HCPRE on the Expression of Oxidative-Stress-Related Genes in *C. elegans*

As shown in Figure 13A, *skn-1* expression in *C. elegans* in the blank control group tended to increase and then decrease over time. Compared with the blank control group, the expression of the *skn-1* gene in *C. elegans* fed 1200 µg/mL HCPRE was significantly increased on both days 4 and 8. As shown in Figure 13B, the expression of the *gcs-1* gene in *C. elegans* in the blank control group gradually decreased over time. On day 4, the expression of the *gcs-1* gene in *C. elegans* fed 800 µg/mL and 1200 µg/mL HCPRE was significantly higher than that in the blank control group. As shown in Figure 13C, the expression of the *gst-4* gene in *C. elegans* in the blank control group gradually increased over time. Compared with the blank control group, the expression of the *gst-4* gene in *C. elegans* fed 800 µg/mL HCPRE was significantly increased on days 4, 8, and 16. As shown in Figure 13D, *gst-7* expression in *C. elegans* in the blank control group gradually decreased over time. On day 4, the expression of the *gst-7* gene in *C. elegans* fed different HCPRE concentrations was significantly higher than that in the blank control group. On day 8, only *C. elegans* fed 1200 µg/mL HCPRE had significantly higher *gst-7* gene expression than the blank control group. On day 16, only *C. elegans* fed 400 µg/mL HCPRE had significantly higher *gst-7* gene expression than the blank control group.

## 3. Discussion

In this study, we extracted polysaccharides of *H. citrina* and determined their extraction yield, content, molecular weight, monosaccharide composition, and in vitro antioxidant properties of HCPRE. In addition, *C. elegans* was used as a model animal and fed different HCPRE concentrations to explore the anti-aging effect of HCPRE.

HCPRE consists of three components with molecular weight distributions of 921,977, 2308, and 200 Da. Monosaccharides mainly include glucose, galactose, galacturonic acid, and mannose. Monosaccharide composition plays an important role in the structural characterization and bioactivity of polysaccharides [26]. Crude polysaccharides from Chinese yam peel and water-soluble polysaccharides from *Pueraria lobata* have also been shown to have DPPH free radical scavenging ability [27,28]. The better ability of HCPRE compared to them may be related to the composition and proportion of the monosaccharides it contains. The antioxidant activity of polysaccharides is related to molecular weight. Zhao et al. compared five different classes of tea polysaccharides and found that all of them possessed antioxidant activity. And the smaller-molecular-weight tea polysaccharides showed stronger antioxidant activity than the larger-molecular-weight tea polysaccharides. This may be due to the fact that polysaccharides with larger molecular weights have higher apparent viscosity and poorer water solubility. As a result, it is difficult for them to pass the tissue barrier into the cell interior, thus limiting their activity [29]. In addition, the structure of polysaccharides has an important effect on their antioxidant activity. The relationship between the structure of HCBPR and its antioxidant activity can be further explored in the future. In addition, the antioxidant activity of polysaccharides is related to their electron- or hydrogen-donating capacity. In this study, HCPRE had higher antioxidant activity than the control group, which may be related to the content of glyoxylate. Glycuronic acid gives HCPRE more electrophilic groups, which can accelerate the release of hydrogen from O-H bonds under acidic conditions [30]. The structure of polysaccharides has an important influence on their antioxidant activity. The structure–activity relationship of HCBPR can be further explored in the future.

The life span of *C. elegans* is the most intuitive physiological indicator of aging. After spawning, *C. elegans* showed shortening of body length and an increase in mortality, and its total spawning volume was negatively correlated with life span [31,32]. Therefore, we added 1% 5-fluorouracil to the NGM medium to prevent *C. elegans* from laying eggs during the oviposition period, thus preventing the effect of egg laying on *C. elegans* life span. Changes in the external environment affect the growth, development, and life span of *C. elegans* [33,34]. Therefore, in this study, we simulated two environments, heat stress and oxidative stress, by increasing the temperature and adding hydrogen peroxide, respectively, to investigate the effects of being fed different concentrations of HCPRE on the life span of *C. elegans* in different environments. Our results showed that being fed 1200 µg/mL HCPRE had a significant positive effect on the life span of *C. elegans* in all three different environments. This suggests that HCPRE not only slows down the aging process of *C. elegans* in normal environments, but also has a protective effect in harsh environments (heat stress and oxidative stress). The insulin/IGF-1 (IIS) signaling pathway is a typical pathway that affects the life span of *C. elegans* [35]. It begins with the binding of insulin-like peptides (ILPs) to *daf-2*. *daf-2* activation triggers a cascade of phosphorylation events through different serine/threonine kinases, such as *age-1*, leading to the phosphorylation of *skn-1* and other transcription factors, which prevents them from translocating to the nucleus and blocks their transcriptional activity, thereby shortening the life span of *C. elegans* [36,37]. Our results showed that HCPRE significantly reduced the expression of *daf-2* and *age-1* genes in *C. elegans* during the early stages of growth. *skn-1* is important for scavenging ROS in *C. elegans* and plays a role in regulating feeding and digestive development during the embryonic stage. From late larval development through early adulthood, *skn-1* can act as a defense against oxidation, delay aging, and maintain normal growth of *C. elegans* [38]. Under specific circumstances, the *skn-1* protein aggregates in cells and activates downstream antioxidant genes such as *gcs-1*, *gst-4*, *gst-7*, and *gst-10* [39]. The *gcs-1* gene activates glutamate-cysteine ligase, which is involved in several processes, including the glutathione biosynthetic process, the response to arsenic-containing substances, and the response to superoxides [40]. The *GST* gene family can activate glutathione transferase and participate in oxidative stress, which regulates life in *C. elegans* [41]. Among these, *gst-4* is related to glutathione metabolism, and *gst-7* is related to innate immunity. They are all homologous to human hematopoietic prostaglandin D synthase gene [42,43]. *gst-10* is orthologous to human glutathione S-transferase pi 1, which plays an important role in the regulation of life span and thermal acclimatization [44]. Increased expression of *GST* genes is one of the defense mechanisms of *C. elegans* and is important for avoiding oxidative damage and prolonging life span [45]. In our study, the expression of *skn-1*, *gcs-1*, *gst-4*, *gst-7*, and *gst-10* genes was all positively correlated with *C. elegans* survival in the HCPRE group. In the pre-growth stage of *C. elegans*, HCPRE may inhibit the oxidative process by upregulating the expression of *skn-1*, *gcs-1*, *gst-4*, and *gst-7* genes, thus prolonging the life span of *C. elegans*. In the middle and later stage of *C. elegans* growth, HCPRE may prevent oxidative damage by upregulating the expression of *gst-4*, *gst-7*, and *gst-10* genes, thus extending the life span of *C. elegans*. In addition, the expression of the *gst-4* gene in *C. elegans* showed a gradual increase over time, indicating that the expression of the *gst-4* gene might be regulated by the increase in ROS in *C. elegans*. The *HSP* gene is related to the expression of heat shock proteins, which can enhance the resistance of the organism to certain environmental stimuli [46]. *Hsp-60* increases the activity of RNA polymerase II, which is mainly involved in the mitochondrial unfolded protein reaction and *C. elegans* larval development [47]. *hsp-16.1* and *hsp-16.2* increase the binding activity of unfolded proteins and are mainly involved in the response to heat [48]. Combined with the results of the genes, HCPRE could significantly increase the gene expression of *hsp-60*, *hsp-16.1*, and *hsp-16.2* on day 8 in all cases. This suggests that HCPRE may increase the resistance of *C. elegans* by upregulating the expression of *hsp-60*, *hsp-16.1*, and *hsp-16.2* genes during the mid-growth period of *C. elegans*, thus prolonging the life span of *C. elegans*.

SOD and CAT are major ROS-scavenging enzymes in vivo and can destroy free radicals, thereby protecting cells [49]. Therefore, we investigated the effects of different concentrations of HCPRE on SOD and CAT activities and the expression of related genes in *C. elegans*. *ctl-1* and *ctl-2* are important CAT coding genes that have protective effects against ROS damage [50]. The *ctl-1* gene is involved in the hydrogen peroxide catabolic process and the response to hydrogen peroxide. It is active in mitochondrion and peroxisome [51]. The *ctl-2* gene is mainly located in the peroxisome and is involved in determining adult life span and peroxisome organization [52]. The inhibition of *ctl-1* and *ctl-2* gene expression has been found to decrease the antioxidant capacity and shorten the life span of *C. elegans* [53]. Our results showed that the expression of both *ctl-1* and *ctl-2* genes was positively correlated with the survival rate of *C. elegans* in the HCPRE group. This suggests that HCPRE may increase CAT activity by upregulating the expression of *ctl-1* and *ctl-2* genes during the pre-growth period of *C. elegans*, thus prolonging its life span. In the middle stages of *C. elegans* growth, HCPRE may increase CAT activity, mainly by upregulating the expression of the *ctl-2* gene, thus slowing the senescence of *C. elegans* and prolonging its life span. *sod-1*, *sod-2*, *sod-3*, and *sod-5* genes are the main genes encoding SOD, which are also capable of protecting the organism from ROS damage. *sod-1* and *sod-5* are directly homologous to human SOD 1 (superoxide dismutase 1), which enhances copper ion-binding activity and superoxide dismutase activity [54,55]. *sod-2* and *sod-3*, directly homologous to human SOD 2 (superoxide dismutase 2), located in the mitochondrial respiratory body, are both involved in the removal of superoxide radicals and can increase protein homodimerization activity and superoxide dismutase activity [56,57]. In the early stage of *C. elegans* growth, HCPRE may increase SOD activity by upregulating the expression of *sod-1* and *sod-3* genes, thus prolonging the life span of *C. elegans*. In the mid-growth stage of *C. elegans*, HCPRE may increase SOD activity by up-regulating the expression of *sod-2* and *sod-3*, thereby extending its life span. In addition, the expression of *sod-1* and *sod-2* genes in *C. elegans* gradually increased over time, indicating that the expression of *sod-1* and *sod-2* genes might be regulated by the increase in ROS in *C. elegans*. In conclusion, HCPRE prolonged the life span of *C. elegans*, with the best results at a concentration of 1200 μg/mL. Compared with the results of *daf-2* and *age-1*, HCPRE had a significant effect on the expression of SOD- and CAT-related genes in *C. elegans* during different growth periods. This suggests that HCPRE prolongs the life span of *C. elegans* mainly by affecting the expression of SOD, CAT, and related genes during different growth periods.

With the increasing aging of the population, anti-aging has attracted widespread attention. Chemical drugs often have uncontrollable risks in anti-aging and are often accompanied by drawbacks such as side effects and drug resistance. Therefore, the search for natural products that can antioxidize or prolong life span has become a hot research topic in related fields, which is of great significance to people. Aging is a complex biological process, and oxidative stress is considered to be the main mechanism limiting life span [58]. In this study, we demonstrated that HCPRE has good antioxidant activity and prolonged the life span of *C. elegans* by regulating antioxidant enzymes, as well as related genes. This provides an experimental basis for further investigation of the anti-aging effects of HCPRE on humans in the future and also promotes the development and research of the *H. citrina* series of pharmaceutical products and functional foods.

## 4. Materials and Methods

### 4.1. Reagents

Petroleum ether, n-butanol, and glucose were purchased from the Beijing Chemical Factory (Beijing, China). Anhydrous ethanol and anhydrous methanol were purchased from Modern Oriental Science and Technology Development Company (Beijing, China). Sodium hydroxide, ferric trichloride, dipotassium hydrogen phosphate, potassium dihydrogen phosphate, and potassium chloride were purchased from Xilong Chemical Company (Shanghai, China). Sulfuric acid and trichloromethane were purchased from Kunshan Jincheng Reagent Company (Suzhou, China). Agar, yeast dipping powder, and tryptone (bio grade) were purchased from Lablead Reagent Company (Beijing, China). The CAT assay kit, SOD assay kit, and ROS assay kit were purchased from Biomed Reagent Company (Beijing, China). The RNA extraction kit, reverse transcription kit, and fluorescence quantitative polymerase chain reaction (PCR) kit were purchased from Novozymes Reagent Company (Beijing, China). Monosaccharide standards mannose, ribose, rhamnose, glucuronic acid, galacturonic acid, N-acetyl-glucosamine, glucose, *N*-acetyl-galactosamine, galactose, xylose, arabinose, and fucose were purchased from Macklin (Shanghai, China).

### 4.2. Preparation of HCPRE

The buds of *H. citrina* were from Datong, Shanxi Province, China. Polysaccharides extraction from *H. citrina* buds was carried out by the ultrasound-assisted water extraction–alcohol precipitation method. *H. citrina* was dried and ground into powder, passed through a 50-mesh sieve, defatted in petroleum ether at 80 °C for 3 h, and then refluxed with 80% ethanol at 85 °C for 6 h. The filter residue was air-dried. Weighing 10 g of filter residue, distilled water was added in the ratio of liquid–solid 20:1 at room temperature, ultrasonicated twice, and the liquid was combined. The ultrasonic power was set at 160 W for 10 min. Then, the liquid was immersed in hot water at 80 °C for 2 h. The process was repeated once, and the extracts were combined. The extract was concentrated under reduced pressure to about 100 mL, and proteins were precipitated by adding about 4 times the volume of anhydrous ethanol. It was placed in a refrigerator at 4 °C overnight. The solid was collected and freeze-dried for use.

### 4.3. Determination of the Extraction Yield and Physicochemical Properties of HCPRE [59]

#### 4.3.1. Determination of Extraction Yield of HCPRE

Determination of the extraction yield of HCPRE by the phenol–sulfuric acid method. Glucose standard solutions of 0.005, 0.01, 0.015, 0.02, 0.025, 0.03, and 0.035 mg/mL and an HCPRE sample solution of 0.01 mg/mL are prepared (dissolved in distilled water). Take 4.0 mL of different concentrations of glucose standard solution and HCPRE sample solution respectively, add 1 mL of 5% phenol solution and 5 mL of concentrated sulfuric acid sequentially and place them in a constant-temperature water bath at 35 °C to heat for 40 min. Cool to room temperature. The absorbance was measured at 490 nm using distilled water as a blank control. A standard curve was established with concentration as the horizontal coordinate and absorbance at 490 nm as the vertical coordinate. The polysaccharide concentration in HCPRE was obtained from the absorbance of the HCPRE sample solution. This test was performed in triplicate.
(1)$$\mathrm{HCPRE}\;\mathrm{rate}\;(\%)=\frac{X\times Y}{Z\times M}\times 100\%$$
(2)$$\mathrm{HCPRE}\;\mathrm{polysaccharide}\;\mathrm{ratio}\;(\%)=\frac{X}{Z}\times 100\%$$
where X is the concentration of sugar contained in the sample solution (g/mL), Y is the total mass of the polysaccharide sample (g), Z is the concentration of the sample solution (g/mL), and M is the mass of defatted and de-sugared cauliflower (g). HCPRE rate represents the percentage of HCPRE from a certain mass of *H. citrina* under this experimental method, and the HCPRE polysaccharide ratio represents the percentage of polysaccharides in the HCPRE obtained under this experimental method.

#### 4.3.2. Determination of Protein Content of HCPRE

The protein content of HCPRE was determined by using Coomassie blue staining. Ww weighed 10 mg of Coomassie brilliant blue G-250, added 5 mL of 95% ethanol, then added 10 mL of 85% phosphoric acid, fixed it to 100 mL with distilled water, and placed it in a brown bottle for spare. Standard solutions of bovine serum albumin at 0.01, 0.02, 0.03, 0.04, 0.05, 0.06, 0.07, 0.08, 0.09, and 0.1 mg/mL and an HCPRE sample solution at 1 mg/mL were prepared (dissolved in distilled water). Different concentrations of bovine serum albumin standard solution and HCPRE sample solution were taken, respectively, and 5 mL of Khao Maas Brilliant Blue G-250 solution was added, mixed well, and then left to stand for 10 min. The absorbance was measured at 595 nm with distilled water as blank control. A standard curve was established with concentration as the horizontal coordinate and absorbance at 595 nm as the vertical coordinate. The protein content in HCPRE was obtained from the absorbance of the HCPRE sample solution. This test was performed in triplicate.

#### 4.3.3. Determination of Total Phenolic Acid Content of HCPRE

The total phenolic acid content of HCPRE was determined by the Folin–Ciocalteu method. The gallic acid standard solutions of 20, 80, 120, 150, 200, 300, and 400 μg/mL and an HCPRE sample solution of 1 mg/mL are prepared (dissolved in distilled water). Pipette 100 μL of gallic acid standard solution and HCPRE sample solution, respectively, add 400 μL of distilled water, shake well, and then add 100 μL of folinol and react for 6 min. Add 1 mL of 10% Na_2_CO_3_ solution and 1 mL of distilled water and stand in the dark for 60 min. The absorbance was measured at 760 nm with distilled water as a blank control. A standard curve was established with concentration as the horizontal coordinate and absorbance at 760 nm as the vertical coordinate. The total phenolic acid content in HCPRE was obtained from the absorbance of the HCPRE sample solution. This test was performed in triplicate.

#### 4.3.4. Determination of Glucuronide Content of HCPRE

The content of HCPRE galacturonic acid is determined by the sulfuric acid–carbazole method. Weigh 0.478 g of sodium tetraborate dissolved in 100 mL of concentrated sulfuric acid. Carbazole 75 mg is weighed precisely and dissolved in 50 mL of anhydrous ethanol to prepare a 0.15% carbazole solution. Standard solutions of galacturonic acid at 20, 40, 60, 80, and 100 μg/mL and a sample solution of HCPRE at 1 mg/mL are prepared (dissolved in distilled water). Pipette 1 mL of galacturonic acid standard solution and HCPRE sample solution, respectively, add 5 mL of sodium tetraborate–sulfuric acid solution and mix well, then heat in a boiling water bath for 20 min. Remove and immediately cool to room temperature, add 0.2 mL of 0.15% carbazole ethanol solution, shake well, and then keep at room temperature for a 20 min standby. The absorbance was measured at 530 nm with distilled water as a blank control. A standard curve was established with concentration as the horizontal coordinate and absorbance at 530 nm as the vertical coordinate. The glucuronide content in HCPRE was obtained from the absorbance of the HCPRE sample solution. This test was performed in triplicate.

#### 4.3.5. Starch Assay for HCPRE

The iodine–potassium iodide method was used to determine whether HCPRE contains starch or not; 2 mg of an HCPRE sample was placed in a test tube and 2 times water was added, then heated for 1 min, cooled, and the iodine–potassium iodide solution was added dropwise, and the color change was observed. This test was performed in triplicate.

#### 4.3.6. Reducing Sugar Assay for HCPRE

The Fehling reagent method is used to determine whether HCPRE contains reducing sugars. Weigh 50 g of sodium hydroxide and 173 g of potassium sodium tartrate dissolved in 500 mL of distilled water (Solution A). Weigh 34.5 g of copper sulfate dissolved in 500 mL of distilled water (Solution B). Prepare a 1% (mass fraction) solution of an HCPRE sample. Add 1 mL of solution A and solution B to the test tube, shake well, and add 4 drops of HCPRE sample solution; heat in a boiling water bath for 3 min., remove and cool, and observe the precipitation and color change. This test was performed in triplicate.

#### 4.3.7. Determination of Molecular Weight of HCPRE

Chromatograph: GPC-20A gel permeation chromatograph; pump: LC20 high-performance liquid chromatography pump; column: TSKgel GMPWXL aqueous gel chromatography column; injector: manual six-way valve injector; detector: differential refractive index (DIRI) detector; mobile phase: 100% isocratic elution of 0.1 N NaNO_3_ + 0.06% NaN_3_ aqueous solution 0.1 N NaNO_3_ + 0.06% NaN_3_ aqueous solution; standard: pullulan polysaccharides with narrow molecular weight distribution; flow rate: 0.6 mL/min; column temperature: 35 °C; injection volume: 20 μL flow rate: 0.6 mL/min; column temperature: 35 °C; injection volume: 20 μL.

#### 4.3.8. Determination of the Monosaccharide Fraction of HCPRE

Hydrolysis: the sample was weighed into a 5 mL ampoule, and the tube was closed by adding 2.0 mL of 2 mol/L trifluoroacetic acid. Acid digestion was carried out at 110 °C for 8 h. After removal, the TFA was evaporated and redissolved in 2.0 mL of water.

Chromatograph: LC-20AD gel permeation chromatograph; column: Xtimate C18 column; injector: manual six-way valve injector; detection wavelength: 250 nm; mobile phase: 0.05 M potassium dihydrogen phosphate solution (adjusted PH to 6.70 with sodium hydroxide solution)—acetonitrile; flow rate: 1.0 mL/min; column temperature: 30 °C; injection volume: 20 μL.

#### 4.3.9. Determination of Antioxidant Capacity of HCPRE In Vitro [60]

Glucose was used as a negative control for the comparison of antioxidant activity.

##### Measurement of 1,1-Diphenyl-2-picrylhydrazyl (DPPH) Radical Scavenging Activity

HCPRE sample solutions of 200, 400, 600, 800, 1000, and 1200 μg/mL are prepared (dissolved in distilled water). Take 0.5 mL of HCPRE samples of different concentrations, add 0.5 mL of 0.04 mg/mL DPPH–methanol solution, and leave it at room temperature for 40 min with sufficient shaking; then measure the absorbance value at 517 nm. Then, take 0.5 mL of HCPRE samples of different concentrations, add 0.5 mL of 0.04 mg/mL DPPH–methanol solution, and do the same. This test was performed in triplicate.
(3)$$\mathrm{DPPH}\;\mathrm{free}\;\mathrm{radical}\;\mathrm{scavenging}\;\mathrm{rate}\;(\%)=\left(1-\frac{A_{3}-A_{2}}{A_{1}}\right)\times 100\%$$
where A_1_ represents the absorbance value with anhydrous methanol, A_2_ represents the absorbance value with DPPH–methanol solution, and A_3_ represents the absorbance value of the HCPRE sample solution.

##### Measurement of Total Reducing Power

Determination of total reducing power by the potassium ferricyanide reduction method: HCPRE sample solutions of 200, 400, 600, 800, 1000, and 1200 μg/mL are prepared. Take 0.5 mL of HCPRE sample solution with different concentrations, add 2.5 mL of 0.2 mol/L pH 6.6 phosphate buffer solution and 2.5 mL of 1% potassium ferricyanide solution, mix well, and keep warm in a water bath at 50 °C for 20 min. Add 2.5 mL of 10% trichloroacetic acid solution and mix well with vibration. After standing for ten minutes take 2.5 mL and add 2 mL of distilled water and 0.5 mL of 0.1% FeCl_3_ solution. Let it stand for 10 min and measure the absorbance value at 700 nm. This test was performed in triplicate.

### 4.4. Culture of E. coli

*E. coli OP 50* was provided by the Dillon Laboratory, Rutgers University, USA. The cryopreserved bacterial solution was placed in a foam box containing ice and allowed to thaw. The *E. coli* suspension was placed on an ultra-clean bench, streaked on LB solid medium, and placed in a biochemical incubator at 37 °C overnight. Single colonies of OP 50 from the delineated culture were isolated into LB liquid medium and incubated overnight in a shaking incubator at 37 °C 220 r/min and stored at 4 °C.

### 4.5. Culture and Treatment of C. elegans

Wild-type *C. elegans* was obtained courtesy of the CGC (Caenorhabditis Genetics Center), USA. *C. elegans* was cultured using nematode growth medium (NGM), fed with *E. coli* OP 50 as food, and incubated at 22 °C in a constant-temperature incubator. HCPRE solutions of 400 µg/mL, 800 µg/mL, and 1200 µg/mL were added to OP 50 bacterial solution at 2% as an experimental group, 400 µg/mL of xylitol polysaccharide as a positive control group, and OP 50 bacterial solution without HCPRE as a blank control group.

### 4.6. Length, Spawning Capacity, and Longevity Measurements

The body length and number of progenies were observed under a stereomicroscope. The day of *C. elegans* contemporization was recorded as d 0. L1-stage *C. elegans* were picked onto NGMs of *E. coli* OP 50 supplemented with different concentrations of HCPRE, and when *C. elegans* had grown to the L4 stage, each *C. elegans* was placed on a separate culture plate. This *C. elegans* was transferred to a new NGM every day, and its length and number of offspring were measured until the end of its spawning. This test was performed in triplicate with 10 *C. elegans* per group.

For life span, *C. elegans* grown to the L1 stage were transferred to different NGMs, and OP 50 bacterial solution supplemented with different concentrations of HCPRE was added to the NGMs, respectively. When they grew to the L4 stage, they were transferred to a medium supplemented with 1% 5-fluorouracil, and the number of *C. elegans* surviving was counted daily until they all died. This test was performed in triplicate with 20 *C. elegans* per group.
(4)$$\mathrm{Life}\;\mathrm{expectancy}\;(d)=\frac{\sum T}{D}$$
(5)$$\mathrm{Relative}\;\mathrm{life}\;\mathrm{change}\;\mathrm{rate}\;(\%)=\frac{S-K}{K}\times 100\%$$
where T is the number of days (d) for each nematode to survive, D is the total number of nematodes, S is the average life span of the experimental group (d), and K is the average life span of the blank control group (d).

### 4.7. Acute Heat Stress Experiment

*C. elegans* cultured to the L4 stage were picked onto a medium containing different concentrations of HCPRE and incubated at 20 °C for 4 d; 1% 5-fluorouracil was added to the medium to block the development of the L4 offspring, and then, the culture was continued at 35 °C for a further period of 2 h. The number of surviving *C. elegans* was observed and counted every 2 h until all of them were dead. This test had 60 *C. elegans* per group.

### 4.8. Acute Oxidative Stress Experiment

*C. elegans* cultured to the L4 stage were picked up and incubated in a medium containing different concentrations of HCPRE at 20 °C for 4 d. The medium was supplemented with 1% 5-fluorouracil to block the development of the offspring at the L4 stage and then washed three times with M9 buffer and transferred to NGM medium containing 1% 5-fluorouracil and 3% H_2_O_2_ at 20 °C. The number of *C. elegans* surviving was recorded every 1 h until all of them were dead. This test had 60 *C. elegans* per group.

### 4.9. Measurement of ROS Levels and CAT and SOD Activities In Vivo

*C. elegans* grown to the L1 stage was transferred to different NGMs, and *E. coli* OP 50 with different concentrations of HCPRE was added to the NGMs and then transferred to a medium supplemented with 1% 5-fluorouracil and incubated at 20 °C. The culture was then incubated at a constant temperature. *C. elegans* grown to the 4th, 8th, and 16th d were washed down with M9 buffer and centrifuged at 1000 rpm for 3 min, and ROS, CAT, and SOD were measured according to the experimental steps in the instruction manual of the ROS, CAT, and SOD assay kits for *C. elegans*, respectively. This test was performed in triplicate.

### 4.10. Quantitative Reverse Transcription-PCR

*C. elegans* grown to the L1 stage was transferred to different NGMs, and OP 50 bacterial solution with different concentrations of HCPRE was added to the NGMs, respectively. When they grew to the L4 stage, they were transferred to the medium supplemented with 1% 5-fluorouracil and continued to be cultured at 20 °C at a constant temperature. *C. elegans* grown to the 4th, 8th, and 16th d were washed down with M9 buffer and centrifuged at 1000 rpm for 3 min. RNA was extracted from *C. elegans* worms according to the instructions of the RNA extraction kit, and reverse transcription and quantitative PCR were performed according to the instructions of the reverse transcription kit and the fluorescent quantitative PCR kit. The internal reference gene for this experiment was *act-1*, which was designed using Primer 6 Primer Design Software. The primer sequences of related genes are shown in Table 5. This test was performed in triplicate.

### 4.11. Statistical Analysis

All results are expressed as mean ± SD. Data were analyzed statistically using SPSS 17.0 software, and LSD and Duncan’s test analyses in the ANOVA module were used to compare differences between groups. *p* < 0.05 was considered significant.

## Figures and Tables

**Figure 1 ijms-25-00655-f001:**
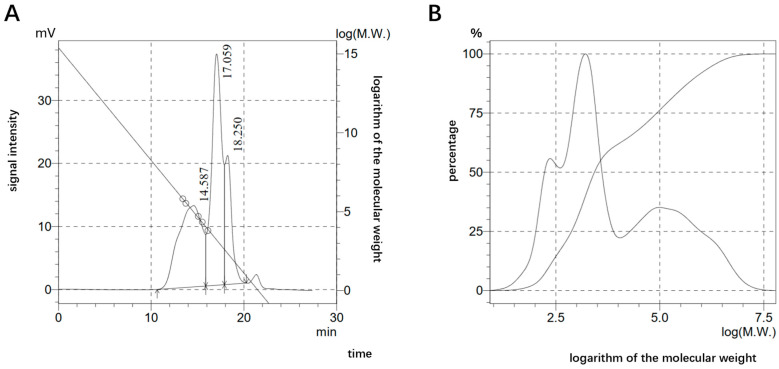
Molecular weight distribution of HCPRE. (**A**) Chromatogram of HCPRE and calibration curve. The diagonal slash is the standard curve. (**B**) The molecular weight distribution curve of HCPRE. Smoothed curves are cumulative distribution plots.

**Figure 2 ijms-25-00655-f002:**
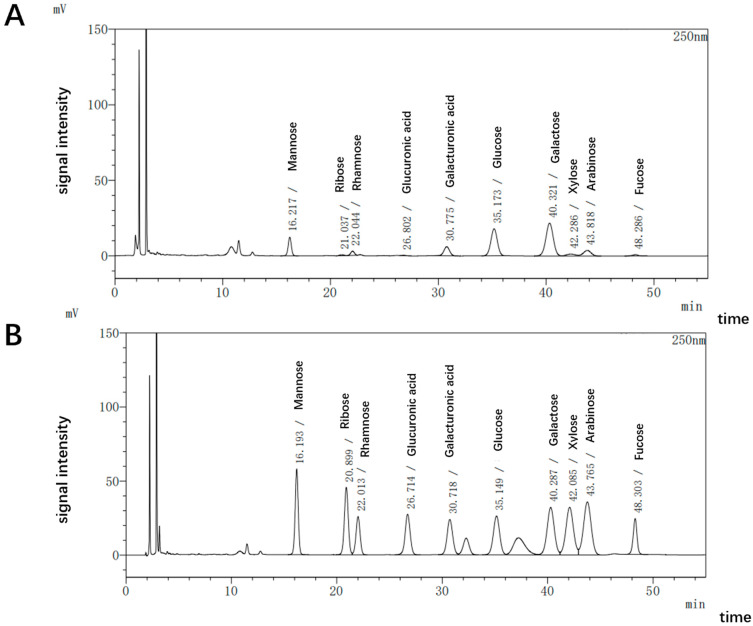
Monosaccharide composition of HCPRE. (**A**) Chromatogram of fungus polysaccharide. (**B**) Chromatogram of 12 standard reference substances.

**Figure 3 ijms-25-00655-f003:**
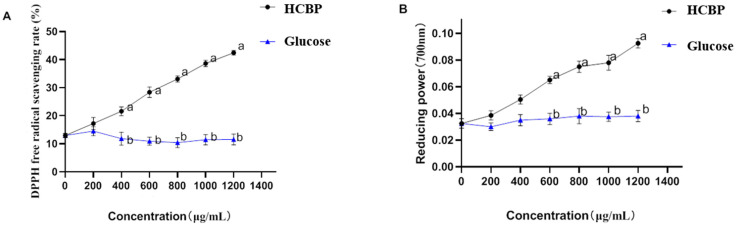
Antioxidant properties of HCPRE in vitro. (**A**) DPPH free radical scavenging curve of HCB. (**B**) Total reducing power curve of HCPRE. Values with different letters are significantly different at *p* < 0.05 (*n* = 3).

**Figure 4 ijms-25-00655-f004:**
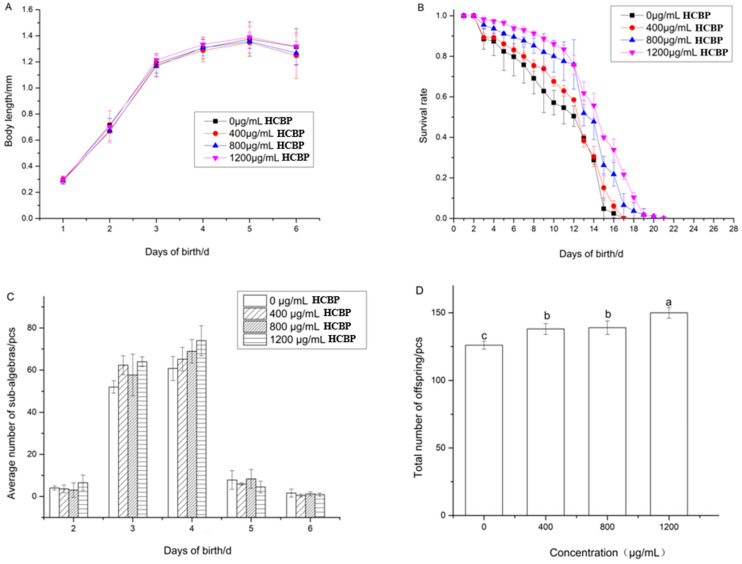
Effects of HCPRE on body length, spawning capacity, and longevity of *C. elegans*. (**A**) Length of *C. elegans* under different concentrations of HCPRE (*n* = 10). (**B**) Survival rate of *C. elegans* under different concentrations of HCPRE (*n* = 20). (**C**) The average progeny of *C. elegans* under different concentrations of HCPRE (*n* = 10). (**D**) The total progeny of *C. elegans* under different concentrations of HCPRE (*n* = 10). Values with different letters are significantly different at *p* < 0.05.

**Figure 5 ijms-25-00655-f005:**
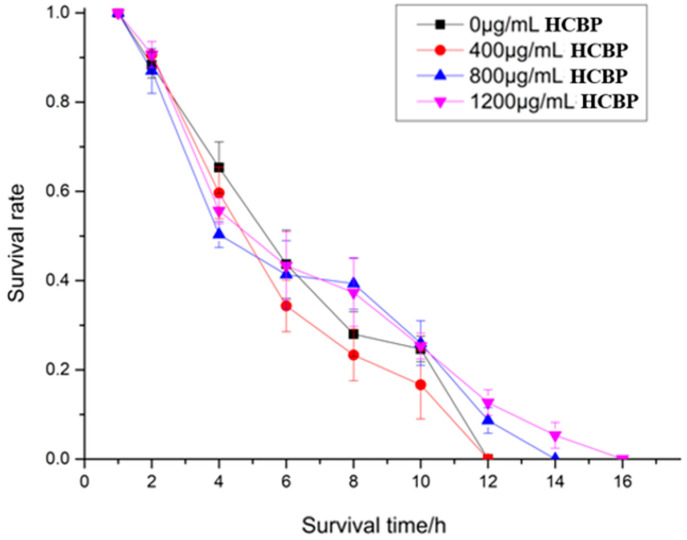
Survival of *C. elegans* under different concentrations of HCPRE under acute thermal stress.

**Figure 6 ijms-25-00655-f006:**
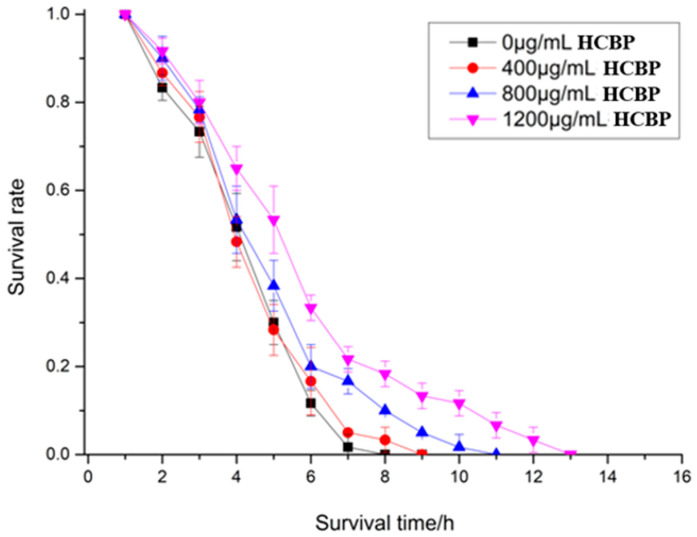
Survival of *C. elegans* under different concentrations of HCPRE under acute oxidative stress.

**Figure 7 ijms-25-00655-f007:**
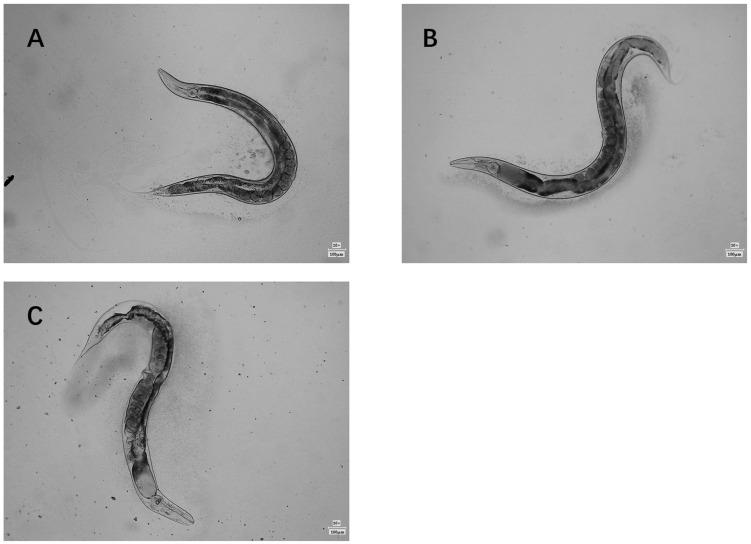
Growth pattern of *C. elegans* in the blank control group at different periods. (**A**) *C. elegans* morphology of the blank control group on day 4. (**B**) *C. elegans* morphology of the blank control group on day 8. (**C**) *C. elegans* morphology of the blank control group on day 16.

**Figure 8 ijms-25-00655-f008:**
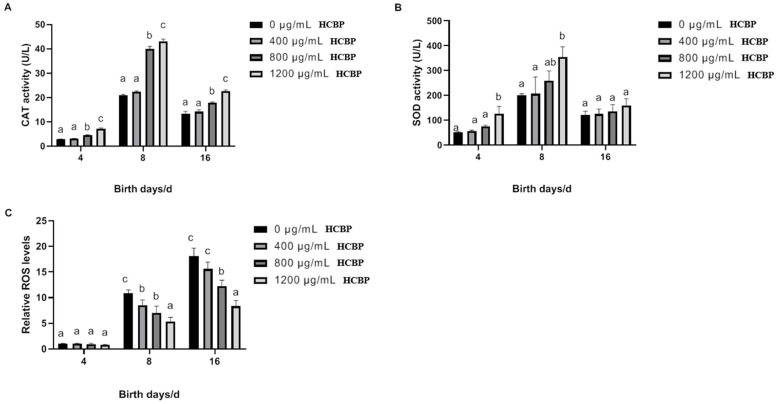
Effect of HCPRE on antioxidant properties of *C. elegans* in vivo. (**A**) Effects of HCPRE on CAT activity in *C. elegans*. (**B**) Effects of HCPRE on SOD activity in *C. elegans*. (**C**) Effects of HCPRE on ROS levels in *C. elegans.* The gene expression of the blank control group on d 4 was used as the relative value, and the four treatment groups for each day were one group. Values with different letters are significantly different at *p* < 0.05 (*n* = 3).

**Figure 9 ijms-25-00655-f009:**
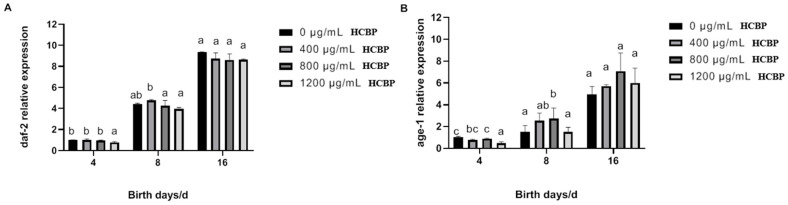
Effect of HCPRE on the expression of longevity-related genes in *C. elegans*. (**A**) Changes in *daf-2* gene in *C. elegans* fed with HCPRE. (**B**) Changes in *age-1* gene in *C. elegans* fed with HCPRE. The gene expression of the blank control group on day 4 was used as the relative value, and the four treatment groups for each day were one group. Values with different letters are significantly different at *p* < 0.05 (*n* = 3).

**Figure 10 ijms-25-00655-f010:**
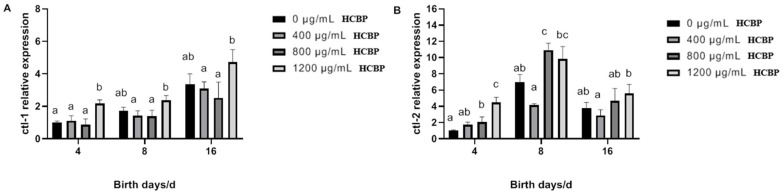
Effect of HCPRE on the expression of CAT-related genes in *C. elegans*. (**A**) Changes in *ctl-1* gene in *C. elegans* fed with HCPRE. (**B**) Changes in *ctl-2* gene in *C. elegans* fed with HCPRE. The gene expression of the blank control group on d 4 was used as the relative value, and the four treatment groups for each day were one group. Values with different letters are significantly different at *p* < 0.05 (*n* = 3).

**Figure 11 ijms-25-00655-f011:**
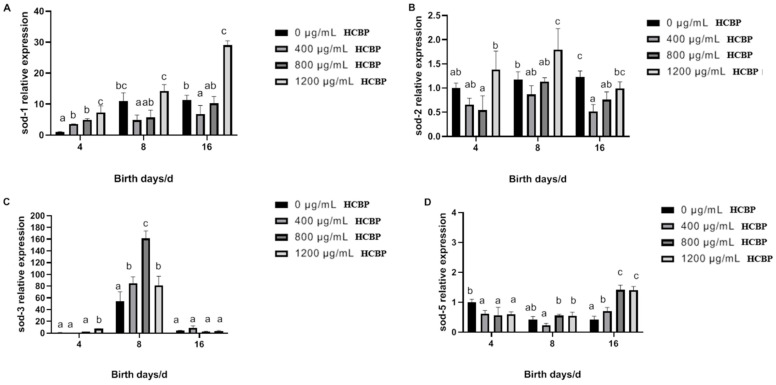
Effect of HCPRE on the expression of SOD-related genes in *C. elegans*. (**A**) Changes in *sod-1* gene in *C. elegans* fed with HCPRE. (**B**) Changes in *sod-2* gene in *C. elegans* fed with HCPRE. (**C**) Changes in *sod-3* gene in *C. elegans* fed with HCPRE. (**D**) Changes in *sod-5* gene in *C. elegans* fed with HCPRE. The gene expression of the blank control group on d 4 was used as the relative value, and the four treatment groups for each day were one group. Values with different letters are significantly different at *p* < 0.05 (*n* = 3).

**Figure 12 ijms-25-00655-f012:**
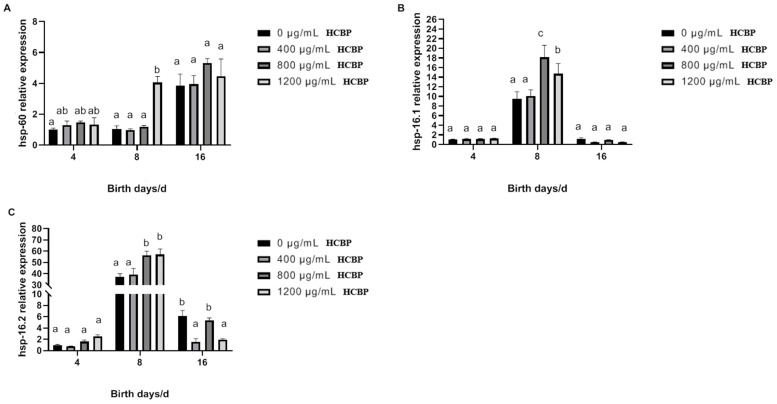
Effect of HCPRE on the expression of thermal stress-related genes in *C. elegans*. (**A**) Changes in *hsp-60* gene in *C. elegans* fed with HCPRE. (**B**) Changes in *hsp-16.1* gene in *C. elegans* fed with HCPRE. (**C**) Changes in *hsp-16.2* gene in *C. elegans* fed with HCPRE. The gene expression of the blank control group on d 4 was used as the relative value, and the four treatment groups for each day were one group. Values with different letters are significantly different at *p* < 0.05 (*n* = 3).

**Figure 13 ijms-25-00655-f013:**
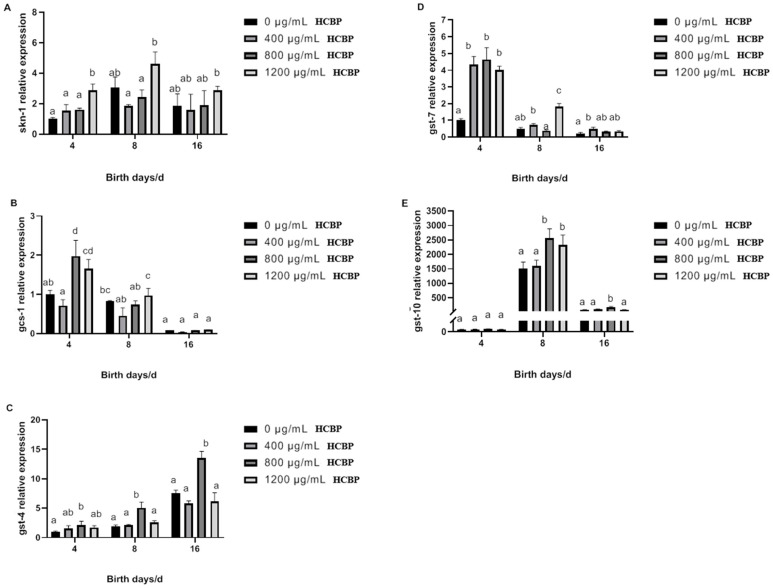
Effect of HCPRE on the expression of oxidative-stress-related genes in *C. elegans*. (**A**) Changes in *skn-1* gene in *C. elegans* fed with p HCPRE. (**B**) Changes in *gcs-1* gene in *C. elegans* fed with HCPRE. (**C**) Changes in *gst-4* gene in *C. elegans* fed with HCPRE. (**D**) Changes in *gst-7* gene in *C. elegans* fed with HCPRE. (**E**) Changes in *gst-10* gene in *C. elegans* fed with HCPRE. The gene expression of the blank control group on d 4 was used as the relative value, and the four treatment groups for each day were one group. Values with different letters are significantly different at *p* < 0.05 (*n* = 3).

**Table 1 ijms-25-00655-t001:** Monosaccharide content of polysaccharide extract of HCPRE.

Number	Name	Relative Content (%)
1	Glucose	31.43
2	Galactose	31.33
3	Galacturonic acid	12.02
4	Mannose	10.08
5	Arabinose	4.80
6	Rhamnose	4.42
7	Fucose	2.68
8	Xylose	2.15
9	Ribose	0.75
10	Glucuronic acid	0.36

**Table 2 ijms-25-00655-t002:** Average life span of *C. elegans* under different concentrations of HCPRE (*n* = 20). Values with different letters are significantly different at *p* < 0.05.

Concentration (μg/mL)	Average Life Span (Day)	Relative Life Change Rate (%)
0	10.78 ± 0.14 ^a^	
400	11.57 ± 0.38 ^a^	7.33
800	13.32 ± 0.87 ^b^	23.56
1200	14.30 ± 1.01 ^b^	32.65

**Table 3 ijms-25-00655-t003:** Average life span of *C. elegans* under different concentrations of HCPRE under acute heat stress (*n* = 60). Values with different letters are significantly different at *p* < 0.05.

Concentration (μg/mL)	Average Life Span (Day)	Relative Life Change Rate (%)
0	7.17 ± 0.18 ^ab^	
400	6.80 ± 0.31 ^a^	−5.16
800	7.89 ± 0.68 ^bc^	10.04
1200	8.44 ± 0.82 ^c^	17.71

**Table 4 ijms-25-00655-t004:** Average life span of *C. elegans* under different concentrations of HCPRE under acute oxidative stress (*n* = 60). Values with different letters are significantly different at *p* < 0.05.

Concentration (μg/mL)	Average Life Span (Day)	Relative Life Change Rate (%)
0	4.51 ± 0.22 ^a^	
400	4.65 ± 0.37 ^a^	3.10
800	5.13 ± 0.62 ^ab^	13.74
1200	5.98 ± 0.83 ^b^	32.59

**Table 5 ijms-25-00655-t005:** Target gene primer sequence.

Gene	Upstream Primer (5′→3′)	Downstream Primer (5′→3′)
*act-1*	CCGTGTTCCCATCCATTGTC	CCAGATCTTCTCCATATCATCCCAG
*daf-2*	GCTCTCGGAACAACCACTGA	GACAAGTCGAAGCCGTCTCA
*age-1*	GACGGAACTCCCGACGTATC	CCCCACTTCATCGGAGCAAT
*skn-1*	GCGACGAGACGAGACGATAA	GAGGTGTTGGACGATGGTGA
*ctl-1*	TCGTTCATGCCAAGGGAGC	GATTCTCCAGCGACCGTTGA
*ctl-2*	TCCCAGATGGGTACCGTCAT	GGTCCGAAGAGGCAAGTTGA
*sod-1*	CTCACTCAGGTCTCCAACGC	AAGTGTGGACCGGCAGAAAT
*sod-2*	TGCCACTTGGTATGAGCCAG	GGCCAGCTTCCAATACCACT
*sod-3*	ACGTGGACAAGGTGGACATC	TTCGCTTTGCTCCAAAAGGC
*sod-5*	GTGGAACTGCTGTCTTCGGA	GCAGACGTACATCCATCGGT
*gcs-1*	AGGTGAATGCGATGCTTGGA	CGATGAGACCTCCGTAAGGC
*gst-4*	CTCTTGCTGAGCCAATCCGT	GCAGTTTTTCCAGCGAGTCC
*gst-7*	GGGAGGAGGCTCAAGTCAAC	CCAGCCGACTTGAGGAAGTT
*gst-10*	ACATTCGGTTCGACTACGAGG	TTCACTAGAGCCTCCGGGAT
*hsp-60*	CCAAGGACGTCAAGTTCGGA	TCCAGCCTCCTCATTAGCCT
*hsp-16.1*	GTCTCGCAGTTCAAGCCAGA	TCGCTTCCTTCTTTGGTGCT
*hsp-16.2*	GTCCAGCTCAACGTTCCGT	CTTGGATTGATAGCGTACGACC

## Data Availability

Data are contained within the article.

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
