# Peer review of "Anti-Aging Effect of *Hemerocallis citrina* Baroni Polysaccharide-Rich Extract on *Caenorhabditis elegans"

_ijms, 2024, doi:10.3390/ijms25010655_

Round 1

Reviewer 1 Report

Comments and Suggestions for Authors

The article "Anti-Aging Effect of Hemerocallis citrina Baroni Polysaccharide Extract on Caenorhabditis Elegans" is an interesting manuscript. The authors prepared HCBP extract, determined its content, and evaluated the biological activity using in vitro and in vivo (C. elegans) methods. The results are well described and discussed. The topic addressed is scientifically interesting. However, I have some suggestions which could improve the manuscript. My comments are below: 

1. In the abstract it is lacking the method's short description

2. The whole name of the plant should be used once in the abstract, and then the short Latin name (H. citrina) should be used (the same comment concerns the main text of the manuscript)

3. Catalase, Superoxide Dismutase, and Reactive oxygen should not be in italics

4. The part extracted from the plant is not mentioned.

5. I suggest that part of the keywords will be different from the title

6. C. elegans (line 49) - /Escherichia coli (line 50) in italic

7. The aim of the study (from line 54...) needs to be more accurate and must be corrected.

8. table 1 presents the content of monosaccharides, but the SD needs to be introduced. Please add it.

9. The methodology needs to contain information on replications of the experiments/measurements. Please, do the authors indicate the n =?

10. The methodology of "Determination of the monosaccharide fraction of HCBP" and "point 4.2.7. should be enriched in information about the ratio used in the mobile phase and if it is a gradient or isocratic method.  

11. Pueraria Lobata should be wroten: Pueraria lobata

12. Could the results obtained be useful for further investigation of the HCBP extract, which could be useful for humans? The authors did not mention it, neither in a discussion nor in the conclusion. Only the Introduction can indicate that it could be interesting for the anti-aging process in humans.

Author Response

Thank you very much for taking the time to review this manuscript.

Reviewer 2 Report

Comments and Suggestions for Authors

Totle, line 2 "Hemerocallis citrina Baroni Polysaccharide Extract". The extract does not contain only polysaccharides but other metabolites as well (as the authors have demonstrated themselves -e.g. proteins, phenols- by the results of their experiments). Furthermore, the name of the botanical authority should not be part of the extract's name and abbreviation further in the text. Therefore,  the extract should be called ""Hemerocallis citrina Polysaccharide-rich Extract". Please change in the title and elsewhere. Also, the abbreviation (abstract line 14 and elsewhere) should be changed to HCPRE.
Line 3, title: Caenorhabditis Elegans please correct to "Caenorhabditis elegans"
Line 11: "Baroni" is the botanical authority so it shouldn't be written in italic. Please correct elsewhere in the text as well.
Line 14: "extraction rate" please change to "extraction yield" here and elsewhere in the text.
Line 37 "Hemerocallis citrina Baroni" on second occurrence, and all the other occurrences further in the text, abbreviate to "H. citrina"
Lines 41-42 "One study showed that 1 mg/mL of Hemerocallis citrina Baroni polysaccharide scavenges 92.56% and 90.68% of superoxide anion and hydroxyl radicals, respectively[16]." The numbers in this sentence are meaningless without the proper context (e.g. type of extracts, concentrations of the extract and the reagents in the assay). Please change to "Extracts of  Hemerocallis citrina Baroni rich in polysaccharides were shown to be scavengers of superoxide anion and hydroxyl radicals[16]. 42"
Line 43 "2,2'-Azinobi-(3-" change to "2,2'-azinobi-(3-"
Line 49 "C. elegans" please write in italic, also correct elsewhere
Line 50 "Escherichia coli" please write in italic, also correct elsewhere if applicable
Line 66 "comprising 921977, 2308, and 200 Da" please change to "comprising 921977 Da, 2308 Da, and 200 Da"
Fig. 1 caption "please write the titles and the units of the coordinates in the figure itself. Not in the caption. The same should apply to the other figures.
Results: please make it clear and mention and comment only on significant differences between the verum and the control group.
Line 281 "Pueraria Lobata" change to "Pueraria lobata"
Discussion: please expand a little with possible chemical mechanisms involving structure-activity relationships, or at least add more examples of similar poylsaccharides exhibiting comparable activity.

Comments on the Quality of English Language

Please check how to write binomial names.

Please check https://www.enago.com/academy/how-to-write-scientific-names-in-a-research-paper-animals-plants/ or some other website or book.

Author Response

Thank you very much for taking the time to review this manuscript.

Please see the attchment.
